# From Silence to Sound: Towards Audio-Visual Subject Customization

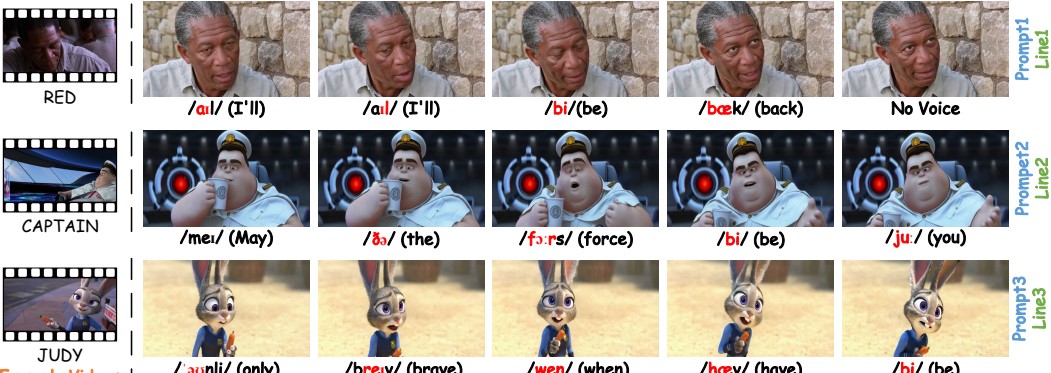

Figure 1: **Showcases produced by our** `VauCustom`. We propose a novel audio-visual subject customization task that generates personalized voice-synchronized videos. `VauCustom` achieves natural audio, high visual fidelity, and accurate synchronization across diverse scenarios, including the real human (top), the animated human (middle), and the animated animal (bottom).

## Abstract

We introduce a novel audio-visual subject customization task that generates videos featuring user-defined characters, emphasizing both visual and audio dimensions. A key challenge is mitigating the gap between visual synthesis and audio learning. To tackle this, we propose VauCustom (**V**ideo-**Au**dio **Custom**), a two-stage method that leverages zero-shot text-to-speech to create personalized audio, and then conditions video synthesis on this audio to unify audio and visuals. During training, we design a decoupled audio-visual learning strategy that models character appearance independently before joint training, thereby preserving the visual fidelity of pre-trained text-to-video models. In addition, we propose a local classifier-free guidance mechanism tailored for audio, which selectively emphasizes character regions based on cross-attention similarity, enhancing audio-visual synchronization while reducing the impact on irrelevant background regions. Experiments demonstrate that VauCustom delivers consistent character appearance, natural audio quality, and precise audio-video synchronization across diverse scenarios, including real humans, animated human characters, and animal characters. We will release all data, code, and models to support future research.

## 1 Introduction

Customized video generation, which aims to create videos featuring user-defined characters, holds significant potential across diverse domains such as film production, animation, and digital humans.

Despite recent progress (Ruiz et al., 2023; Wang et al., 2024; Wu et al., 2025; Hu et al., 2025), existing works predominantly emphasize the visual dimension—generating character appearances and movements—while neglecting a vital component that makes videos truly expressive: synchronized speech, including both natural audio and lip movements. This situation parallels the silent film era, where the lack of sound limited the immersive power of the medium.

Incorporating audio with visuals can greatly enhance realism and engagement. Several directions have explored this. *Portrait animation methods* (Shen et al., 2023; Zhang et al., 2023b; Sun et al., 2023; Cui et al., 2024) achieve impressive audio-driven facial motions but typically require an initial video frame and synchronized audio, limiting their use when such inputs are unavailable. *Joint audio-visual generation models*, such as JavisDiT (Liu et al., 2025a) and MMDiffusion (Ruan et al., 2023), attempt end-to-end generation from text but struggle with limited capacity and data. More recently, Veo3 (GoogleDeepMind, 2025) showed the potential to generate high-quality audiovisual content directly from text, but it remains closed source and does not support personalized character customization, which is important for content creators and digital assistants.

These efforts highlight both opportunities and gaps in the field. A practical solution must simultaneously achieve (1) faithful character customization, (2) natural and synchronized audio-visual generation, and (3) accessible research resources, including an open benchmark to support systematic evaluation. To this end, we introduce the **audio-visual subject customization** task, which extends customized video generation from silent visuals to synchronized audiovisual outputs. As illustrated in Fig. 1, users define a character and provide prompts and dialogue lines, and the system generates expressive audiovisual videos across diverse contexts. To facilitate research on this task, we construct a comprehensive benchmark covering real humans, animated human characters, and animal-style characters, along with standardized evaluation protocols on three dimensions: character appearance, audio quality, and audio-visual synchronization.

To address this new task, we propose `VauCustom`, a two-stage framework for audio-visual subject customization. In the first stage, we leverage state-of-the-art zero-shot text-to-speech models (Du et al., 2024; An et al., 2024) to synthesize personalized character audio with high naturalness and speaker consistency. In the second stage, video generation is conditioned on this audio, unifying visual synthesis and audio. However, existing open-source video generation models (Ho et al., 2022; Chen et al., 2023; Yang et al., 2024; Wan et al., 2025) are trained without audio inputs and often suffer from appearance degradation when extended to the audio-visual setting. To address this, we design a **decoupled audio-visual learning strategy**, which first models character appearance independently and then jointly incorporates audio conditioning. This strategy preserves the visual fidelity of pretrained text-to-video models while enabling robust audio-driven guidance.

Furthermore, we enhance synchronization through a **local classifier-free guidance (CFG) mechanism** tailored for audio conditions (Ho & Salimans, 2022). Since spoken dialogue mainly affects the character rather than the background, we identify character regions using cross-attention similarity between text and video features, and apply audio-based CFG selectively to them. This localized conditioning improves lip–audio alignment while avoiding degradation of visual quality that may arise if audio signals indiscriminately influence the entire frame. Together, these components enable `VauCustom` to achieve high-fidelity, well-synchronized audiovisual outputs.

Extensive experiments validate the effectiveness of `VauCustom`, showing consistent improvements in appearance fidelity, audio quality, and audio–visual synchronization across a variety of scenarios. This work establishes the first systematic benchmark for audio-visual subject customization and advances model design in this domain, laying a foundation for future research on expressive and personalized digital media. In summary, our contributions are threefold:

- We define the **audio-visual subject customization** task and establish the first benchmark, featuring diverse subjects such as real humans, animated humans, and animal-style characters, along with standardized metrics for appearance, audio quality, and synchronization.

- We propose `VauCustom`, a two-stage framework integrating zero-shot TTS audio with audio-conditioned video synthesis to achieve synchronized audiovisual customization.

- We introduce tailored strategies, including a **decoupled audio-visual learning** method to preserve visual fidelity and a **region-selective audio CFG mechanism** to improve lip–audio synchronization, leading to superior results across diverse scenarios.

## 2 RELATED WORK

**Video diffusion models.** Inspired by the success of image diffusion (Ho et al., 2020; Rombach et al., 2022), many attempts have extended diffusion models to the video domain. Early approaches adapted pretrained text-to-image models to the temporal domain, as shown by Make-A-Video (Singer et al., 2022) and MagicVideo (Zhou et al., 2022), which synthesized motion by processing latent representations across time. AnimateDiff (Guo et al., 2023) introduces motion adapters that preserve spatial details while producing temporally coherent motion. More recently, diffusion transformer (DiT) architectures (Peebles & Xie, 2023) have gained prominence, with models like CogVideoX (Yang et al., 2024) and WAN (Wan et al., 2025). These large-scale frameworks (Chen et al., 2023; Yang et al., 2024; Kong et al., 2024; Wan et al., 2025) show remarkable capabilities in generating high-quality content through extensive training on video-text pairs.

Going further, some works explore the joint generation of video and synchronized audio. For instance, MoCha (Wei et al., 2025) generates character video from speech input, while JarvisDiT (Liu et al., 2025a) simultaneously generates both video and speech. However, these methods do not support customizing both the character's visual identity and voice, a gap our work addresses.

**Portrait Image Animation.** The task of portrait image animation involves generating a talking face video by animating a single portrait image based on a driving audio signal. Pioneering works such as SadTalker (Zhang et al., 2023b) utilized 3D Morphable Models (Egger et al., 2020) to convert speech into facial motion coefficients, addressing key challenges in lip sync and head movement. In parallel, other methods leveraged 2D representations. For instance, AniPortrait (Wei et al., 2024) used facial landmarks as an intermediate representation to steer the synthesis process. A significant trend has been the integration of diffusion models to enhance visual quality and generalization. Frameworks like DiffTalk (Shen et al., 2023) and VividTalk (Sun et al., 2023), for example, leverage diffusion backbones to produce high-fidelity results. Building on this foundation, works like EMO (Tian et al., 2024) further focus on capturing rich facial expressions. The Hallo series (Xu et al., 2024; Cui et al., 2025; 2024) extends these capabilities by addressing long-duration synthesis and incorporating pretrained transformer-based video models to achieve more vivid generation. Despite recent advances, existing methods typically assume access to both an initial video frame of the target character and a corresponding audio input. This assumption limits their applicability in scenarios with scarce visual assets, restricting broader creative and practical use.

**Customized Generation.** Customized generation aims to preserve user-specified subject identities and can be grouped into two categories. *Instance-specific customization* adapts identity preservation through per-subject fine-tuning. Methods such as Still-Moving (Chefer et al., 2024), CustomCrafter (Wu et al., 2025), and Video Alchemist (Chen et al., 2025a) extend identity learning to temporal sequences by repeating input frames and embedding subject features. Multi-subject systems like CustomVideo (Wang et al., 2024) and DisenStudio (Chen et al., 2024) further localize identities using attention maps and subject masks. While effective, these approaches require separate optimization for each identity. *End-to-end customization* encodes identity information via conditioning networks, eliminating the need for per-subject training. For faces, ID-Animator (He et al., 2024) employs facial adapters with identity-preserving losses, while ConsisID (Yuan et al., 2024) leverages dual-frequency analysis for richer representation. Recent methods such as ConceptMaster (Huang et al., 2025), Phantom (Liu et al., 2025b), and SkyReels-A2 (Fei et al., 2025) further advance multi-subject customization through stronger text–image binding.

While progress has been made, most approaches focus primarily on visual elements, overlooking a vital component that brings videos to life—character speech, including both lip movements and synchronized audio. In this work, we explicitly address this overlooked aspect.

## 3 METHOD

### 3.1 TASK DEFINITION

We propose a novel task termed audio-visual subject customization, which generates synchronized video and audio from textual inputs, guided by a small set of exemplars. In the customization phase, the model receives exemplars, each containing a video $\nu^{\text{ref}} \in \mathbb{R}^{T \times H \times W \times 3}$, its aligned audio $\alpha^{\text{ref}} \in \mathbb{R}^{T \times D}$, a descriptive prompt $c^{\text{ref}}$, and a spoken line $l^{\text{ref}}$. These exemplars capture both

Figure 2: **Inference pipeline of** `VauCustom`. Given a text prompt and a dialogue line, our method produces a personalized video with synchronized audio in two stages: (a) customized speech is synthesized using reference audio of the character, and (b) a personalized video is generated in alignment with the synthesized audio, conditioned on both the input text prompt and the audio.

the character's visual traits (appearance, motion patterns) and audio traits (voice characteristics, speaking style). At test time, given a new prompt $c$ describing the visual context and a dialogue line $l$, the model generates a video $\nu$ consistent with the character's traits and matching $c$, along with synchronized audio $\alpha$ that renders $l$ in the character's voice. Formally,

$$\nu, \alpha = \mathcal{F}(c, l; \{\nu^{\text{ref}}, \alpha^{\text{ref}}, c^{\text{ref}}, l^{\text{ref}}\}), \tag{1}$$

where $\mathcal{F}$ maps text inputs to aligned video-audio outputs with exemplar guidance.

## 3.2 OVERALL PIPELINE

To address audio-visual subject customization, we design a two-stage pipeline that factorizes the function $\mathcal{F}$ into two sequential components, as shown in Fig. 2. This separation enables dedicated handling of audio and video customization while ensuring tight alignment.

In the first stage, the audio customization function $\mathcal{F}_1$ generates personalized speech. Given a dialogue line $l$, a prompt $c$, and reference audio exemplars $\alpha^{\text{ref}}$, $\mathcal{F}_1$ produces a customized audio track $\alpha$ that preserves the speaker's voice and style while adapting to the semantics in $l$:

$$\alpha = \mathcal{F}_1(c, l; \alpha^{\text{ref}}). \tag{2}$$

We employ the open-source CosyVoice2 (Du et al., 2024) for its simplicity and effectiveness in zero-shot text-to-speech, though other synthesis methods are also compatible (An et al., 2024; Ju et al., 2024; Ding et al., 2025). In the second stage, the video customization function $\mathcal{F}_2$ synthesizes a clip $\nu$ aligned with both the generated audio $\alpha$ and the text prompt $c$, while preserving the character's identity from exemplars $\nu^{\text{ref}}, \alpha^{\text{ref}}, c^{\text{ref}}$:

$$\nu = \mathcal{F}_2(c, \alpha; \nu^{\text{ref}}, \alpha^{\text{ref}}, c^{\text{ref}}). \tag{3}$$

Implementation details of $\mathcal{F}_2$ are presented in the following sections.

## 3.3 DECOUPLED LEARNING FOR AUDIO-CONDITION INJECTION

Given the lack of open-source text-to-video (T2V) models with native audio control, we extend WAN (Wan et al., 2025), a state-of-the-art T2V framework, to support audio-conditioned generation as the second stage of `VauCustom`. WAN progressively decodes video sequences from Gaussian noise with text cross-attention for text prompt conditioning. We build on this foundation by injecting audio conditions while maintaining the original model's visual fidelity.

**Audio conditioning.** To introduce audio conditioning, each attention block is augmented with a randomly initialized audio cross-attention module, which encodes speech-related cues, as shown in Fig. 2b. Queries are drawn from video latents, while keys and values come from audio features $c_\alpha$ extracted using Wav2Vec (Schneider et al., 2019; Baevski et al., 2020). Following ControlNet (Zhang et al., 2023a), we apply zero-initialization to the final FFN layer of the audio

Figure 3: **Illustration of the decoupled audio-visual learning.** Training is staged as follows: (a) learn a dataset-level appearance LoRA on a talking-face dataset, (b) freeze it and train audio cross-attention modules for audio-driven synchronization, (c) learn a customized appearance LoRA from reference videos of the subject of interest, and (d) jointly integrate the customized LoRA with audio cross-attention for synchronized personalized video generation.

cross-attention, ensuring output consistency with the original WAN before training. To preserve high-quality synthesis, the WAN backbone remains frozen, and only the new audio parameters are trained on talking-face datasets (Cui et al., 2024) with a flow-matching objective.

**Decoupled audio-visual learning.** A significant challenge in audio-visual modeling is the tendency of audio conditioning modules to inadvertently learn appearance-specific biases from the talking-face training dataset. These biases can impair the model's ability to generalize to unseen identities. To overcome this, we design a staged decoupled learning strategy, as illustrated in Fig. 3.

In the first stage (Fig. 3a), we learn a dataset-level appearance LoRA on a large-scale talking-face dataset. These adapters capture generic visual traits and motion dynamics, while the audio pathway remains untouched. In the second stage (Fig. 3b), we freeze the appearance LoRA and introduce audio cross-attention modules trained with the audio-video data, allowing the model to focus purely on temporal synchronization from audio cues. In the third stage (Fig. 3c), we build subject-specific personalization by training a customized appearance LoRA on reference video frames of the target character. Finally (Fig. 3d), the dataset-level LoRA from the first stage is discarded, and the customized LoRA is integrated with the trained audio cross-attention modules for joint learning. During joint training, we introduce learnable audio tokens that encode character-dependent voice traits and propagate them through the audio cross-attention module, as shown in Fig. 2b.

This decoupled learning design separates appearance modeling and audio synchronization into distinct yet composable stages. Such modularization enables `VauCustom` to maintain high-fidelity visual identity while achieving precise and natural audio-video alignment.

### 3.4 REGION-SELECTIVE AUDIO CLASSIFIER-FREE GUIDANCE

To further improve audio-visual synchronization in `VauCustom`, we enhance the sampling process from the perspective of CFG, aiming to provide stronger and more targeted audio conditioning.

**Text classifier-free guidance.** CFG is a widely used inference technique in diffusion models. It strengthens the alignment between generated content and input conditions by interpolating between conditional and unconditional predictions. The text-based CFG is formulated as:

$$\tilde{v}_t = \mathbf{w}_{txt} \cdot [v_t(z_t; c) - v_t(z_t; \phi)] + v_t(z_t; \phi), \tag{4}$$

where $c$ is the text condition, $\phi$ indicate empty condition, and $\mathbf{w}_{txt}$ controls the guidance strength.

**Global audio CFG.** We intend to incorporate audio as an additional condition in CFG for improved audio-visual synchronization. A straightforward solution is to assign separate guidance scales to text and audio, as in Loopy (Jiang et al., 2025):

$$\tilde{v}_t = \mathbf{w}_{aud} \cdot [v_t(z_t; c_{aud}, c_{txt}) - v_t(z_t; c_{txt})] + \mathbf{w}_{txt} \cdot [v_t(z_t; c_{txt}) - v_t(z_t; \phi)] + v_t(z_t; \phi). \tag{5}$$

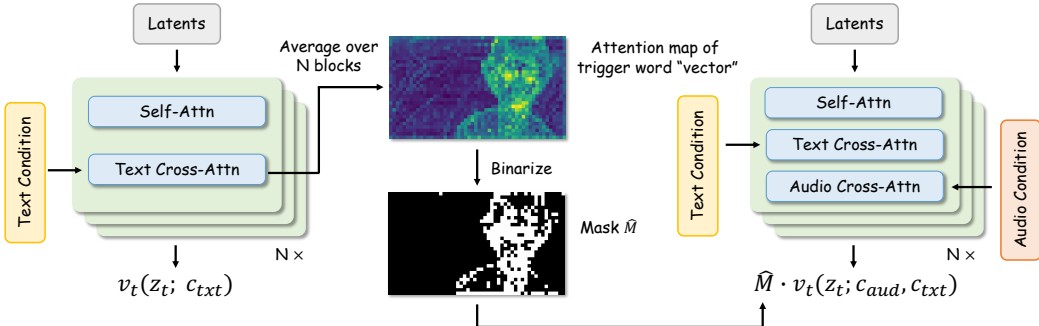

Figure 4: **Region-Selective Audio Classifier-Free Guidance.** A spatial mask $\hat{M}$ is derived from text cross-attention maps of the character's trigger word. The mask localizes audio guidance to character regions, ensuring audio-driven motion while preserving background fidelity.

Here $\mathbf{w}_{aud}$ is the audio guidance weight. However, applying it uniformly across all spatial tokens introduces artifacts and degrades quality, as the audio signal—meant for localized lip and facial motion—erroneously propagates into irrelevant background regions.

**Region-selective audio CFG.** To tackle this, we introduce a regions-selective CFG that restricts audio guidance to character regions, as shown in Fig. 4. We derive a spatial mask $\hat{M}$ from text-to-image cross-attention maps, following the intuition of Prompt2Prompt (Hertz et al., 2023). The attention matrix is written as $M = \text{Softmax}\left(\frac{QK^T}{\sqrt{d}}\right)$, where $M_{ij}$ denotes the influence of the $j$-th text token on the $i$-th spatial location. We average the attention maps across layers for the tokens tied to the character's trigger word—the special prompt token explicitly representing the target character—and threshold them to obtain a binary mask $\hat{M}$, where 1 marks character regions and 0 denotes background. The CFG update then becomes:

$$\tilde{v}_t = \mathbf{w}_{aud} \cdot \hat{M} \cdot [v_t\left(z_t; c_{aud}, c_{txt}\right) - v_t\left(z_t; c_{txt}\right)] + \mathbf{w}_{txt} \cdot [v_t\left(z_t; c_{txt}\right) - v_t\left(z_t; \phi\right)] + v_t\left(z_t; \phi\right). \quad (6)$$

By localizing audio guidance to character regions, this strategy improves audio-video synchronization while preserving background fidelity. Results in Sec. 4.3 show that region-selective audio CFG captures audio-driven motion in the subject without distorting irrelevant areas.

## 4 EXPERIMENTS

### 4.1 EXPERIMENTAL SETUP

**Implementation details.** In the first stage, we adopt CosyVoice2 (Du et al., 2024) for zero-shot TTS. In the second stage, video customization is built on the Wan2.1 14B model (Wan et al., 2025). Our audio cross-attention module mirrors the architecture of text cross-attention in Wan. It is randomly initialized, except for the final layer, which is zero-initialized. For appearance learning, we apply LoRA with rank 128. Training proceeds in several phases: (1) 10K steps for appearance fitting and 6K steps for audio-following on the Hallo3 dataset (Cui et al., 2024); (2) 5K steps of appearance LoRA training; and (3) 3K steps of joint audio-visual training. At inference, we set the audio classifier-free guidance (CFG) weight to $\mathbf{w}_{aud} = 9$ and the text CFG weight to $\mathbf{w}_{txt} = 5$.

**Benchmark and evaluation metrics.** We curated customization data for 60 characters from movies and animations, with about 10 audio-video clips per character for training. Each clip was paired with text annotations generated by Qwen2.5-72B (QwenTeam, 2024). For evaluation, we collected 8 test videos per character, also annotated with Qwen2.5-72B prompts, and used Kimi Audio (Ding et al., 2025) to generate dialogue lines from the audio. In total, the benchmark includes 480 test samples. We evaluate the generated content along three dimensions: video quality, audio quality, and audio-visual synchronization. Details of these metrics are provided in Appendix B.

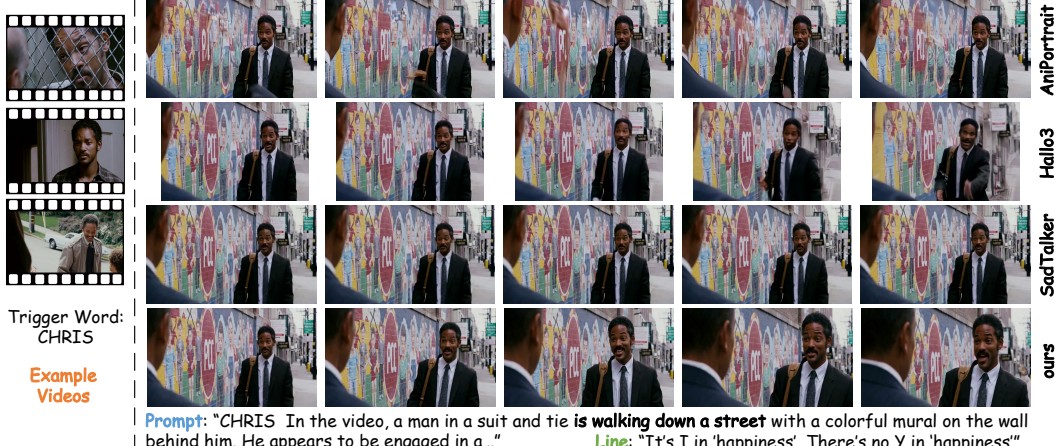

Figure 5: **Qualitative comparisons**. AniPortrait and Hallo3 show facial artifacts, while all baselines yield mostly static backgrounds. In contrast, our method generates dynamic scenes that match the text (*e.g.*, walking down a street) with high visual quality and consistent character appearance.

Table 1: **Quantitative comparisons with prior methods.** Compared to SadTalker, AniPortrait, and Hallo3, our method achieves higher dynamics and better audio–visual synchronization, while maintaining strong visual quality and consistent character appearance.

| Method | Text Clip Similarity↑ | Video Clip Similarity↑ | Subject Dino Similarity↑ | Dynamic Degree↑ | Temporary Consisitency↑ | Sync-D↓ | Sync-C↑ |
|---|---|---|---|---|---|---|---|
| SadTalker (Zhang et al., 2023b) | 0.275 | 0.827 | 0.606 | 0.003 | **0.992** | 9.242 | 2.899 |
| AniPortrait (Wei et al., 2024) | 0.264 | 0.831 | 0.588 | 0.426 | 0.971 | 10.134 | 1.395 |
| Hallo3 (Xu et al., 2024) | 0.269 | 0.839 | 0.615 | 0.279 | 0.982 | 9.120 | 2.908 |
| VauCustom (Ours) | **0.287** | **0.859** | **0.638** | **0.717** | 0.970 | **8.526** | **3.056** |

## 4.2 MAIN RESULTS

Since existing video customization models do not support audio conditioning, we compare our method with the representative portrait animation approaches: Hallo3 (Cui et al., 2024), SadTalker (Zhang et al., 2023b), and AniPortrait (Wei et al., 2024). All require a source image as input, so following MoCha, we provide the first frame of our generated video as their input.

**Qualitative comparisons.** As shown in Fig. 5, AniPortrait and Hallo3 produce noticeable facial artifacts, *e.g.*, blur in the last frame. Moreover, backgrounds in all baselines remain mostly static, limiting motion and expressiveness. In contrast, our method better matches the text description (*e.g.*, walking down a street), generating dynamic scenes with high visual quality that align with audio. More qualitative comparisons are presented in Appendix D.1.

**Quantitative comparisons.** As shown in Table 1, our method achieves state-of-the-art results on four of five primary metrics. It attains the highest Text-CLIP and Video-CLIP Similarity, demonstrating strong semantic alignment with the input text. High Subject-DINO Similarity confirms consistent identity preservation, while a Dynamic Degree of 0.717 reflects more expressive motion. VauCustom also excels in Sync-C and Sync-D, underscoring its advantage in audio-visual synchronization. Higher Sync-C indicates more accurate lip–audio alignment, while lower Sync-D reflects better motion fidelity. The improvements on both metrics highlight the realism and coherence of our generated audio-video.

## 4.3 ABLATION STUDIES

**Ablations on decoupled audio-visual learning.** We ablate the training strategy with two baselines: (i) removing audio pre-training (*w/o pt.*), and (ii) removing the decoupled design, where audio and appearance are jointly optimized (*w/o decoup.*) on the talking-face data. As shown in Fig. 6, without audio pre-training the model captures subject appearance from a few samples but produces

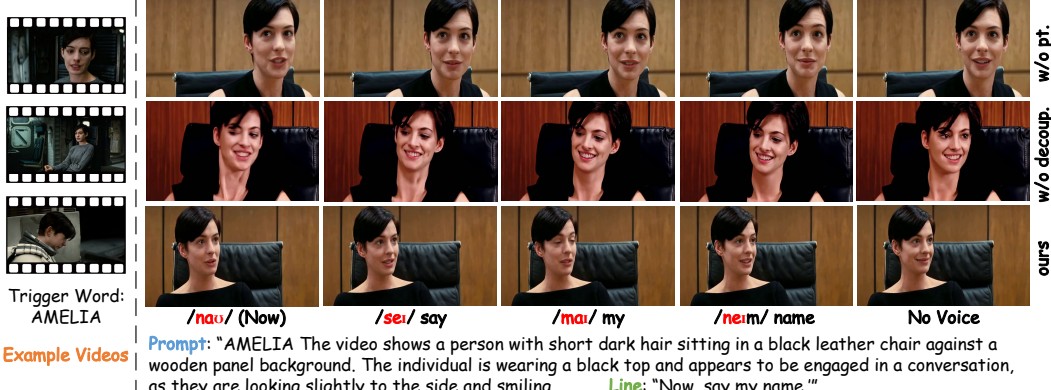

Figure 6: **Qualitative ablations on decoupled learning.** Without audio pre-training (*w/o* pt.), the mouth stays closed despite speech. Without decoupling (*w/o* decoup.), synchronization improves, but appearance fidelity degrades.

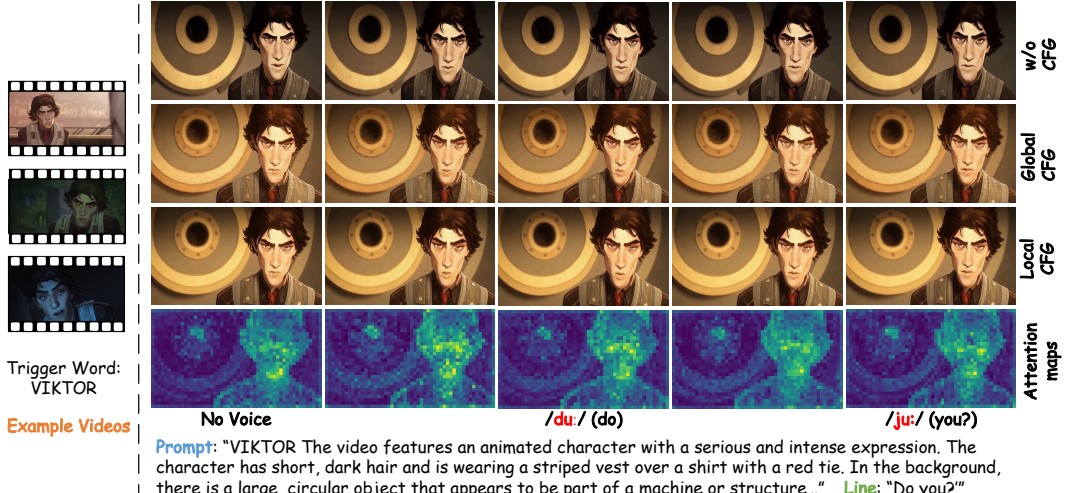

Figure 7: **Ablations on region-selective audio CFG.** Without CFG, synchronization fails—the character opens the mouth even without speech. Global CFG improves synchronization but causes grid-like background artifacts. In contrast, our local CFG delivers coherent, high-quality results with accurate synchronization, as evidenced by attention maps concentrated on the face region.

spurious mouth movements even when no speech occurs. Without decoupling, both audio–visual synchronization and appearance fidelity deteriorate. We suspect the joint optimization amplifies biases in the pre-training distribution. Our full model strikes the best balance, preserving appearance while ensuring audio–visual alignment. Quantitative results in Table 2 confirm this: our method achieves the highest appearance similarity (Subject-DINO) and best synchronization metrics (lowest Sync-D, highest Sync-C), validating the effectiveness of the decoupled learning strategy.

**Ablations on region-selective audio CFG.** We compare three variants: text-only CFG (*w/o* CFG), global audio CFG (Global CFG), and our proposed region-selective audio CFG (Local CFG). As shown in Fig. 7, *w/o* CFG results in poor synchronization—the character opens their mouth even when silent. Global CFG improves synchronization but introduces grid-like background artifacts, likely from strong audio signals affecting irrelevant regions. In contrast, Local CFG balances visual quality and synchronization, producing coherent results. Cross-attention maps confirm strong activations around character regions, providing effective spatial guidance for audio conditioning. Table 2 supports this: while removing audio CFG has little impact on visual quality (*e.g.*, CLIP similarity), it markedly degrades synchronization, with lower Sync-C and higher Sync-D.

Table 2: **Quantitative ablations.** Without decoupling (*w/o* decoup.), synchronization improves slightly, but appearance fidelity drops. Without region-selective audio CFG (*w/o* local CFG), visual quality remains strong but synchronization degrades.

| Method | Text Clip Similarity↑ | Video Clip Similarity↑ | Subject Dino Similarity↑ | Dynamic Degree↑ | Temporary Consisitency↑ | Sync-D↓ | Sync-C↑ |
|---|---|---|---|---|---|---|---|
| *w/o* audio pt | **0.287** | 0.854 | 0.634 | 0.702 | 0.968 | 10.518 | 2.152 |
| *w/o* decoup. | 0.278 | 0.831 | 0.594 | 0.665 | 0.958 | 9.840 | 2.646 |
| *w/o* local CFG | 0.285 | 0.855 | 0.625 | **0.734** | **0.970** | 9.199 | 2.843 |
| VauCustom (Ours) | **0.287** | **0.859** | **0.638** | 0.717 | **0.970** | **8.526** | **3.056** |

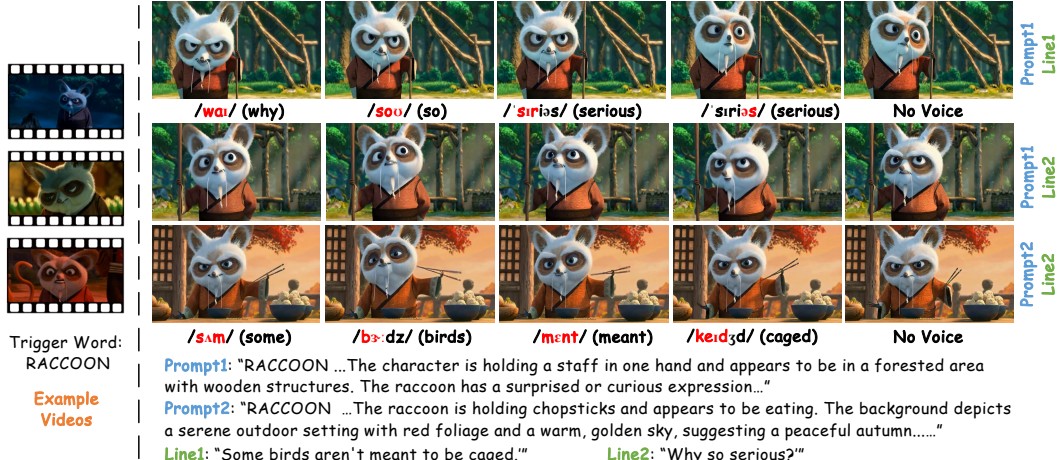

Figure 8: **Flexible audio-visual composition.** The first and third rows show generations with different prompts and speech lines. Comparing rows 1–2 shows consistent visuals across audio, while rows 2–3 show consistent audio across prompts. More examples presented in Appendix D.2.

## 4.4 FLEXIBLE AUDIO-VISUAL COMPOSITION

As shown in Fig. 8, our approach enables flexible audio-visual composition, where text prompts and speech lines can be combined in different ways to produce diverse and controllable videos. When both components are varied simultaneously (1st and 3rd rows), the generated results exhibit distinct scene contexts as well as different lip movements. With a fixed prompt but varied speech (1st and 2nd row), the model preserves a consistent visual appearance while adapting the articulation to the spoken line. Conversely, with a fixed speech input but varied prompts (2nd and 3rd row), the videos present different backgrounds and character contexts while maintaining accurate lip synchronization. These results showcase the flexibility of our approach in integrating text-conditioned visual generation with audio-conditioned dynamics.

## 5 CONCLUSION AND FUTURE WORK

We studied the task of audio-visual subject customization, which aims to generate videos of user-defined characters with consistent visual identity and synchronized speech. To address this challenge, we proposed VauCustom, a two-stage framework that decouples appearance learning from audio-driven synthesis, and introduced a region-selective audio classifier-free guidance mechanism to improve synchronization by focusing on character-relevant regions. While our approach achieves promising results, several challenges remain. Future work includes developing fully end-to-end systems that adapt to unseen characters and diverse speech styles without fine-tuning, and extending the framework to multi-character or conversational settings where identity preservation and cross-speaker synchronization are crucial. We hope this work brings greater attention to audio-visual customization and inspires continued progress toward unified, adaptive, and user-centric generative models for personalized media creation.

## ETHICS STATEMENT

This work focuses on advancing customized audio-visual generation with applications in creative content creation, human–computer interaction, and assistive communication. We are mindful of the potential misuse of personalized audio-visual generation systems, particularly in creating misleading or harmful content such as deepfakes. To mitigate these risks, we restrict our experiments to publicly available datasets from movies and animations, avoiding the use of private or sensitive personal data. Our system is intended solely for research and educational purposes, and we encourage responsible use under ethical guidelines. Furthermore, we advocate for transparent disclosure when synthetic media is used, and for the development of complementary safeguards (*e.g.*, watermarking, detection tools) to reduce risks of misuse.

## REPRODUCIBILITY STATEMENT

We make every effort to ensure the reproducibility of our work. Upon publication, we will release the complete codebase, pretrained models, and training scripts. All datasets used are either publicly available or will be released under appropriate licenses. Comprehensive implementation details, including model architectures, training schedules, and hyperparameter settings, are documented in the main text (see Section 4.1) and Appendix B. By providing these resources, we aim to enable the community to reproduce our experiments faithfully and facilitate further research.

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

# LLM USAGE

In this section, we clarify the role of large language models (LLMs) in preparing this work. The model was used exclusively for language polishing, such as refining grammar, style, and readability, without contributing to the research design, analysis, or conclusions.

## A  APPENDIX OVERVIEW

This appendix provides additional implementation details, empirical analysis, and extended results to supplement the main paper. It is organized as follows:

- **Appendix B: Details on Evaluation Metrics**
  Provides further details on evaluation metrics, including:
  - Appendix B.1: Evaluation Metrics for Visual Quality
  - Appendix B.2: Evaluation Metrics and Results for Audio Quality
  - Appendix B.3: Evaluation Metrics for Audio-Visual Synchronization
- **Appendix C**: Human Evaluation.
- **Appendix D**: More Results, including:
  - Appendix D.1: More Results on Qualitative Comparisons
  - Appendix D.2: More Results on Flexible Audio-Visual Composition
  - Appendix D.3: More Demo Results

## B  DETAILS ON EVALUATION METRICS

### B.1  EVALUATION METRICS FOR VISUAL QUALITY

To comprehensively evaluate our personalized video generation model, we assess the quality of the outputs across multiple dimensions. Following Video Alchemist (Chen et al., 2025b), our evaluation framework incorporates the following five metrics:

- **Text CLIP Similarity**: Measures the semantic alignment between the generated video and the input text. Specifically, we compute the average cosine similarity between CLIP ViT-L/14 (Radford et al., 2021) features of the text prompt and those of each generated frame.
- **Video CLIP Similarity**: Assesses perceptual similarity between the generated video and the ground truth by averaging the cosine similarity between their CLIP ViT-L/14 (Radford et al., 2021) features across all frames.
- **Subject DINO Similarity**: Evaluates how well the target character's identity is preserved. We extract DINO ViT-B/16 (Caron et al., 2021) features from the reference subject image and compare them with features of the segmented character in generated frames, averaging the cosine similarity. Character regions are segmented using Grounding-DINO Swin-T (Liu et al., 2023) and SAM ViT-B/16 (Kirillov et al., 2023).
- **Dynamic Degree**: Quantifies motion magnitude by computing optical flow between adjacent frames with RAFT (Teed & Deng, 2020). Larger flow magnitudes correspond to more pronounced movement.
- **Temporal Consistency**: Following VBench (Huang et al., 2024), measures visual smoothness by calculating the average cosine similarity between CLIP ViT-L/14 (Radford et al., 2021) features of consecutive frames. Higher values indicate greater coherence and fewer flickering artifacts.

### B.2  EVALUATION METRICS AND RESULTS FOR AUDIO QUALITY

We evaluate the generated audio using three standard metrics that capture speaker similarity, transcription accuracy, and speech naturalness. Table S1 reports the quantitative results. The evaluation follows the official ZipVoice implementation[1].

---

[1] https://github.com/k2-fsa/ZipVoice

Table S1: Audio quality evaluation results. Higher is better (↑) for SIM-o and UTMOS, while lower is better (↓) for WER.

| Metric | Score |
|---|---|
| Speaker Similarity (SIM-o) ↑ | 0.508 |
| UTMOS ↑ | 3.45 |
| Seed-TTS WER (weighted) ↓ | 14.32% |

- **Speaker Similarity (SIM-o)**: Measures how closely the generated voice matches the target speaker's reference by computing cosine similarity between speaker embeddings. Following the ZipVoice protocol, we use an ECAPA-TDNN verifier with a fine-tuned WavLM-Large front-end. Higher values indicate stronger similarity.

- **Word Error Rate (WER)**: Computed with the Whisper-large-v3 ASR model, as in Seed-TTS. This reflects the transcription accuracy of generated speech compared to ground-truth text. Lower values are better.

- **UTMOS (Objective MOS)**: Predicted using the UTMOS22Strong model to assess speech naturalness. Evaluations are performed on 16 kHz single-ended audio. Higher scores denote more natural, human-like speech.

### B.3 EVALUATION METRICS FOR AUDIO-VISUAL SYNCHRONIZATION

We evaluate the generated audio-video using two standard metrics that capture audio-visual synchronization, including Sync-C (Synchronization Confidence) and Sync-D (Synchronization Distance)

- **Sync-C** measures the confidence of lip–audio alignment. It is computed using a pretrained lip–audio synchronization model (SyncNet (Chung & Zisserman, 2016)) by evaluating the correlation between mouth-region features and the corresponding audio. Higher values indicate stronger synchrony.

- **Sync-D** quantifies the misalignment error between lip movements and audio features. It captures the temporal discrepancy estimated by the synchronization model, where lower values denote better alignment.

Together, Sync-C and Sync-D provide complementary perspectives on audio–visual synchronization, reflecting both matching strength and alignment accuracy.

## C HUMAN EVALUATION

We conducted a detailed human evaluation to assess the quality of the generated content, focusing on six key criteria that capture different aspects of performance for the audio-visual subject customization task. Each criterion was rated on a 4-point Likert scale, where higher scores indicate better quality: 1 = Not Good, 2 = Borderline Reject, 3 = Borderline Accept, and 4 = Good.

- **Lip-Sync Accuracy**: Evaluates how precisely the character's lip movements match the accompanying audio.

- **Facial Expression Realism**: Assess whether character facial expressions appear natural, contextually appropriate, and free of robotic or exaggerated qualities.

- **Action Naturalness**: Measures how smoothly the character's body movements and gestures align with the spoken audio.

- **Text Alignment**: Examines how well the character's actions and expressions reflect the behaviors and descriptions specified in the input prompt.

- **Subject Alignment**: Determines whether the generated character remains consistent with the identity, appearance, and attributes of the target subject.

Table S2: **Human evaluation results.** Each criterion was rated on a 4-point Likert scale by 20 independent participants. Participants rated each method on six aspects: lip-sync accuracy, facial expression realism, action naturalness, text alignment, subject alignment, and visual quality.

| Method | Lip-Sync Quality | Facial Expression Realism | Action Naturalness | Text Alignment | Subject Alignment | Visual Quality |
|---|---|---|---|---|---|---|
| Hallo3 (Xu et al., 2024) | 2.55 | 2.83 | 2.15 | 2.32 | 3.17 | 2.96 |
| SadTalker (Zhang et al., 2023b) | 2.75 | 2.59 | 2.77 | 2.81 | 3.31 | 3.07 |
| AniPortrait (Wei et al., 2024) | 1.42 | 1.56 | 1.33 | 1.87 | 2.11 | 1.62 |
| VauCustom (Ours) | **3.51** | **3.32** | **3.18** | **3.64** | **3.69** | **3.55** |

- **Visual Quality**: Evaluates the overall visual fidelity of the output, checking for artifacts, discontinuities, or other visual defects.

To ensure robust evaluation, we randomly sampled 50 instances from the test set and recruited 20 independent participants to evaluate each sample, yielding over 1,000 human ratings per model. As summarized in Table S2, our method consistently outperforms all baseline approaches across all six criteria, achieving average scores approaching **3.48**. These results confirm that our approach produces more accurate, realistic, and faithful audio-visual customizations compared to existing methods.

# D  MORE RESULTS

## D.1  MORE RESULTS ON QUALITATIVE COMPARISONS

We provide additional qualitative comparisons with existing baselines in Fig. S1–S4. Across different subjects and scenarios, our method consistently generates videos with fewer facial artifacts and more dynamic backgrounds, while maintaining subject fidelity and alignment with the given input conditions. These results further demonstrate the robustness of our approach in diverse contexts.

## D.2  MORE RESULTS ON FLEXIBLE AUDIO-VISUAL COMPOSITION

We further illustrate the flexibility of our method in combining prompts and speech lines. As shown in Fig. S5 and S6, our approach generates coherent videos from the same prompt while adapting articulation to different speech lines. Conversely, Fig. S7 and S8 show that with the same speech input, the model adapts naturally to different prompts and backgrounds while preserving accurate lip synchronization. These results highlight the compositional controllability of our framework.

## D.3  MORE DEMO RESULTS

Finally, we provide additional demo results of audio-visual subject customization in Fig. S9–Fig. S14, showing consistent identity, natural motion, and accurate synchronization across diverse subjects.

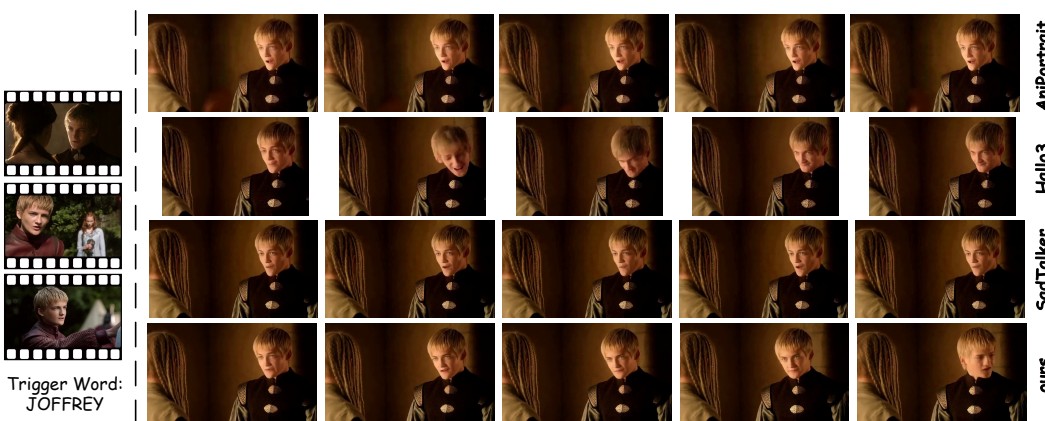

Trigger Word:
JOFFREY

Example Videos

**Prompt**: "JOFFREY The image depicts two characters in a dimly lit room, engaged in a conversation. The character on the right is wearing a dark, ornate tunic with intricate designs and a brooch at the collar. This character has short, light-colored hair and appears to be speaking or reacting to something. The character on the left, whose back is partially turned to the camera, has long, braided hair and is wearing a simple, light-colored garment. The setting suggests a medieval or fantasy context, possibly from a television show or movie. The lighting creates a dramatic atmosphere, highlighting the expressions and details of their attire."

**Line**: "Be queen over all of us."

Figure S1: More qualitative comparison with existing methods.

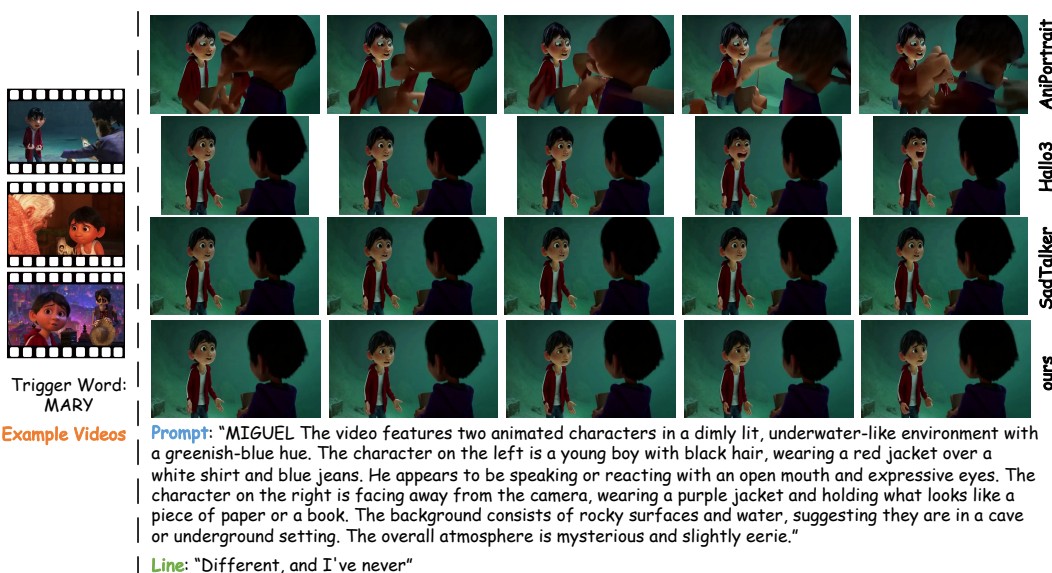

Trigger Word:
MARY

Example Videos

**Prompt**: "MIGUEL The video features two animated characters in a dimly lit, underwater-like environment with a greenish-blue hue. The character on the left is a young boy with black hair, wearing a red jacket over a white shirt and blue jeans. He appears to be speaking or reacting with an open mouth and expressive eyes. The character on the right is facing away from the camera, wearing a purple jacket and holding what looks like a piece of paper or a book. The background consists of rocky surfaces and water, suggesting they are in a cave or underground setting. The overall atmosphere is mysterious and slightly eerie."

**Line**: "Different, and I've never"

Figure S2: More qualitative comparison with existing methods.

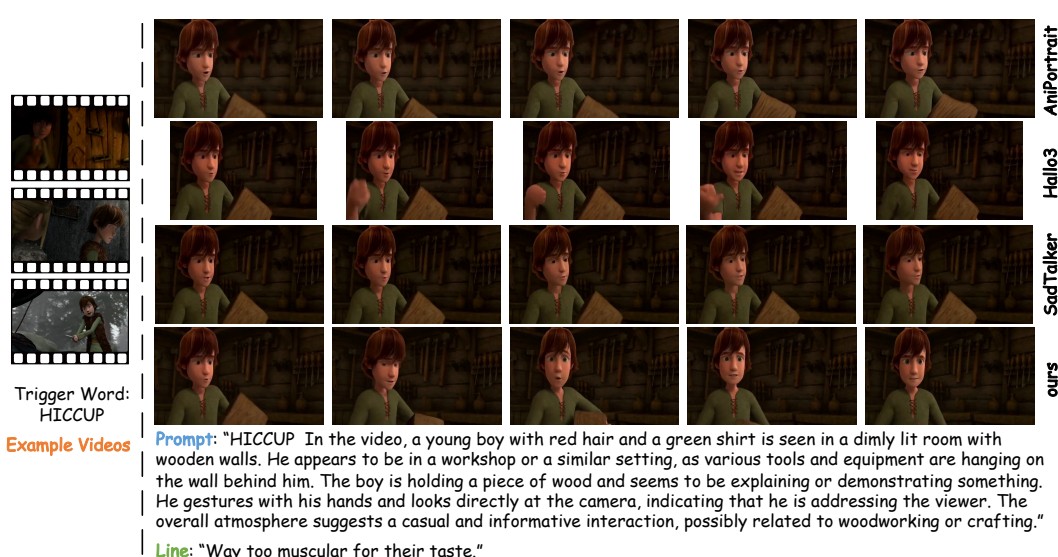

Trigger Word:
ROBARTS

Example Videos

Prompt: "ROBARTS The video depicts a courtroom scene, likely from a classic film or television show. The central figure is an older man dressed in traditional British legal attire, including a black robe and a white wig, indicating he is a judge or barrister. He appears to be speaking, possibly delivering a verdict or addressing the court. Behind him, there are several other individuals, some also wearing legal robes and wigs, suggesting they are fellow judges or barristers. The audience in the background consists of men and women dressed in formal attire, attentively listening to the proceedings. The setting is formal and serious, typical of a courtroom environment.

Line: "You were at 25 minutes past nine."

Figure S3: More qualitative comparison with existing methods.

Trigger Word:
HICCUP

Example Videos

Prompt: "HICCUP In the video, a young boy with red hair and a green shirt is seen in a dimly lit room with wooden walls. He appears to be in a workshop or a similar setting, as various tools and equipment are hanging on the wall behind him. The boy is holding a piece of wood and seems to be explaining or demonstrating something. He gestures with his hands and looks directly at the camera, indicating that he is addressing the viewer. The overall atmosphere suggests a casual and informative interaction, possibly related to woodworking or crafting."

Line: "Way too muscular for their taste."

Figure S4: More qualitative comparison with existing methods.

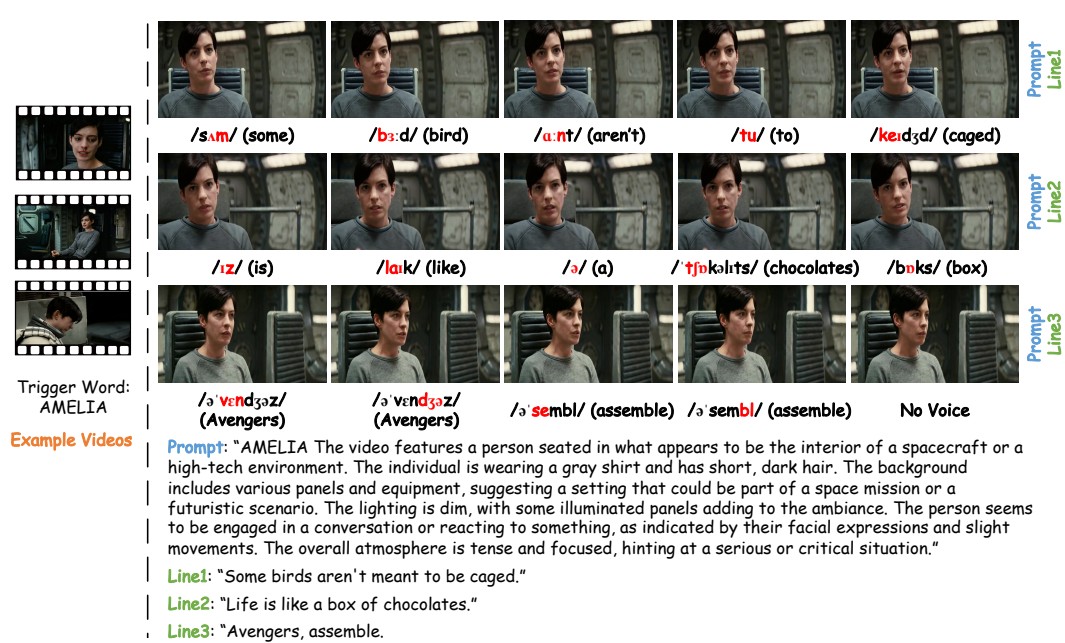

Figure S5: **More results on flexible audio-visual composition.** Our method generates videos from the same prompt while adapting to different speech lines.

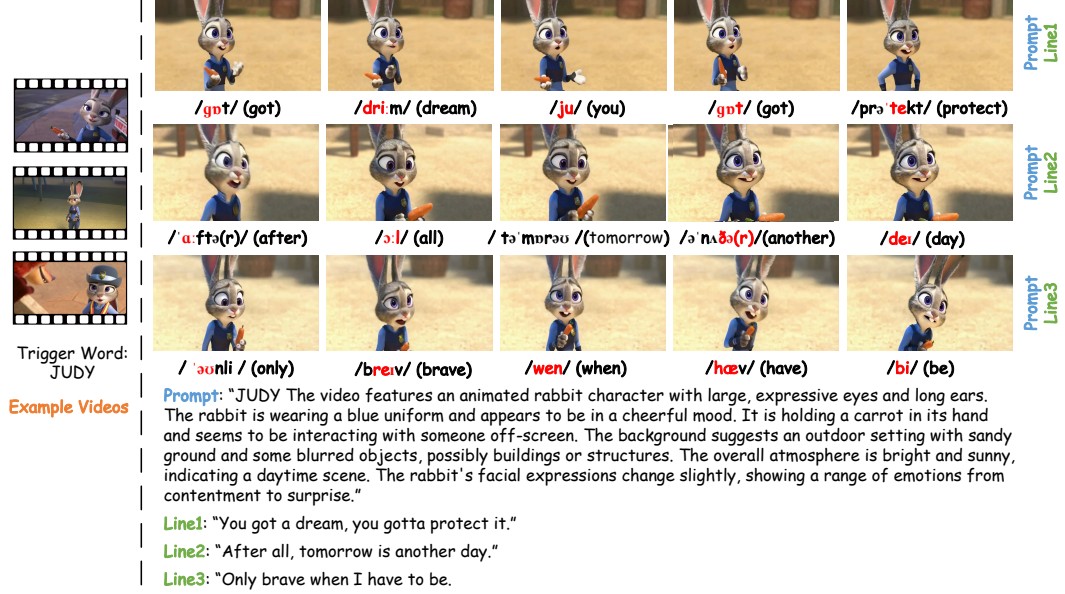

Figure S6: **More results on flexible audio-visual composition.** Our method generates videos from the same prompt while adapting to different speech lines.

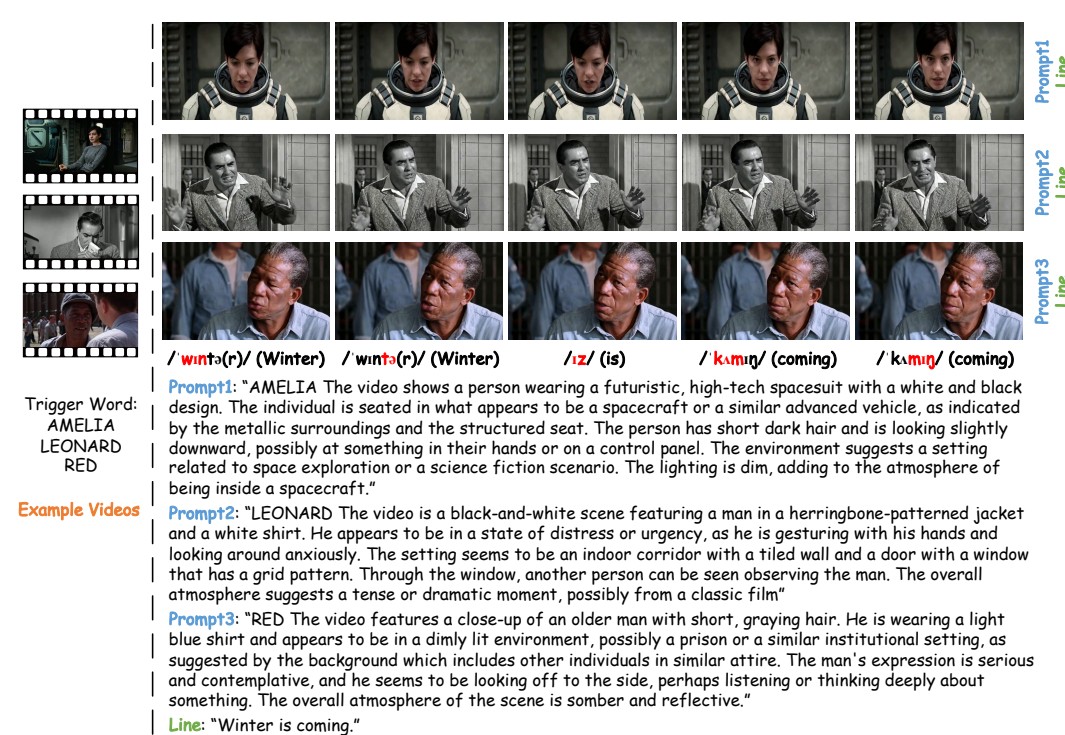

**Trigger Word:**
AMELIA
LEONARD
RED

Example Videos

/ˈwɪntə(r)/ (Winter)  /ˈwɪntə(r)/ (Winter)  /ɪz/ (is)  /ˈkʌmɪŋ/ (coming)  /ˈkʌmɪŋ/ (coming)

**Prompt1**: "AMELIA The video shows a person wearing a futuristic, high-tech spacesuit with a white and black design. The individual is seated in what appears to be a spacecraft or a similar advanced vehicle, as indicated by the metallic surroundings and the structured seat. The person has short dark hair and is looking slightly downward, possibly at something in their hands or on a control panel. The environment suggests a setting related to space exploration or a science fiction scenario. The lighting is dim, adding to the atmosphere of being inside a spacecraft."

**Prompt2**: "LEONARD The video is a black-and-white scene featuring a man in a herringbone-patterned jacket and a white shirt. He appears to be in a state of distress or urgency, as he is gesturing with his hands and looking around anxiously. The setting seems to be an indoor corridor with a tiled wall and a door with a window that has a grid pattern. Through the window, another person can be seen observing the man. The overall atmosphere suggests a tense or dramatic moment, possibly from a classic film"

**Prompt3**: "RED The video features a close-up of an older man with short, graying hair. He is wearing a light blue shirt and appears to be in a dimly lit environment, possibly a prison or a similar institutional setting, as suggested by the background which includes other individuals in similar attire. The man's expression is serious and contemplative, and he seems to be looking off to the side, perhaps listening or thinking deeply about something. The overall atmosphere of the scene is somber and reflective."

**Line**: "Winter is coming."

Figure S7: **More results on flexible audio-visual composition.** Our method generates videos from the same speech lines while adapting to different prompts.

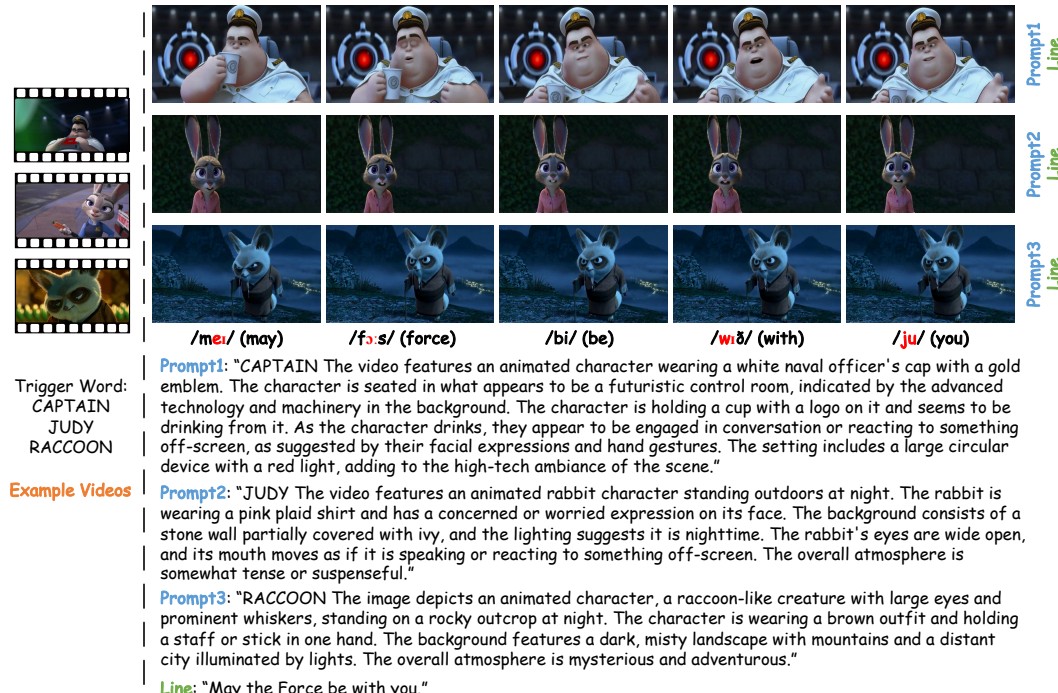

**Trigger Word:**
CAPTAIN
JUDY
RACCOON

Example Videos

/meɪ/ (may)  /fɔːs/ (force)  /bi/ (be)  /wɪð/ (with)  /ju/ (you)

**Prompt1**: "CAPTAIN The video features an animated character wearing a white naval officer's cap with a gold emblem. The character is seated in what appears to be a futuristic control room, indicated by the advanced technology and machinery in the background. The character is holding a cup with a logo on it and seems to be drinking from it. As the character drinks, they appear to be engaged in conversation or reacting to something off-screen, as suggested by their facial expressions and hand gestures. The setting includes a large circular device with a red light, adding to the high-tech ambiance of the scene."

**Prompt2**: "JUDY The video features an animated rabbit character standing outdoors at night. The rabbit is wearing a pink plaid shirt and has a concerned or worried expression on its face. The background consists of a stone wall partially covered with ivy, and the lighting suggests it is nighttime. The rabbit's eyes are wide open, and its mouth moves as if it is speaking or reacting to something off-screen. The overall atmosphere is somewhat tense or suspenseful."

**Prompt3**: "RACCOON The image depicts an animated character, a raccoon-like creature with large eyes and prominent whiskers, standing on a rocky outcrop at night. The character is wearing a brown outfit and holding a staff or stick in one hand. The background features a dark, misty landscape with mountains and a distant city illuminated by lights. The overall atmosphere is mysterious and adventurous."

**Line**: "May the Force be with you."

Figure S8: **More results on flexible audio-visual composition.** Our method generates videos from the same speech lines while adapting to different prompts.

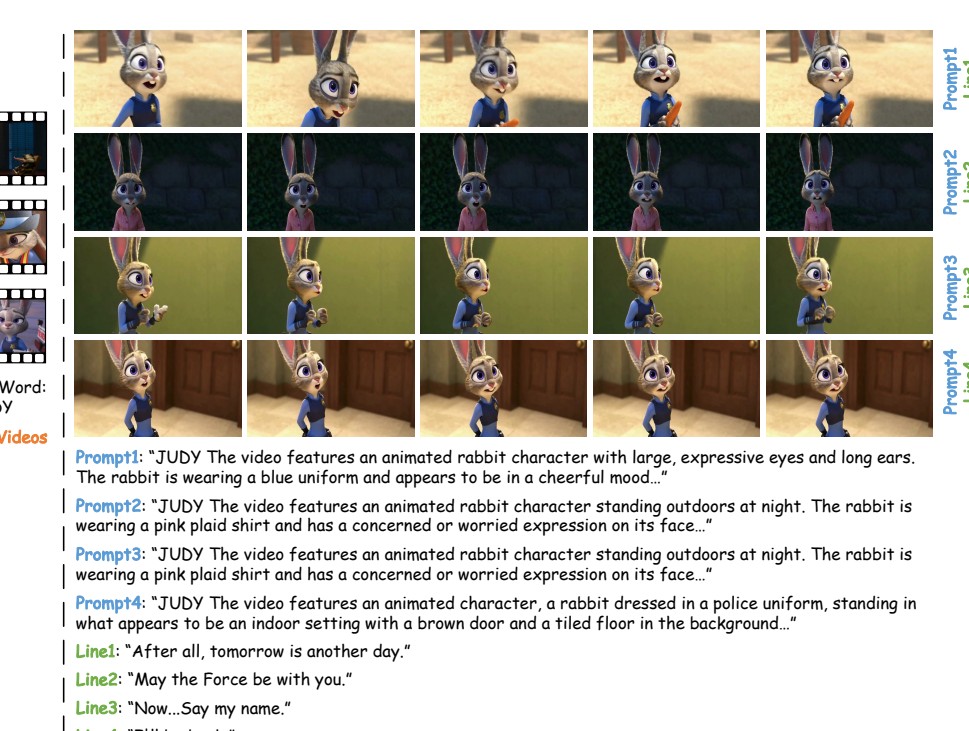

Trigger Word:
JUDY

Example Videos

**Prompt1**: "JUDY The video features an animated rabbit character with large, expressive eyes and long ears. The rabbit is wearing a blue uniform and appears to be in a cheerful mood…"

**Prompt2**: "JUDY The video features an animated rabbit character standing outdoors at night. The rabbit is wearing a pink plaid shirt and has a concerned or worried expression on its face…"

**Prompt3**: "JUDY The video features an animated rabbit character standing outdoors at night. The rabbit is wearing a pink plaid shirt and has a concerned or worried expression on its face…"

**Prompt4**: "JUDY The video features an animated character, a rabbit dressed in a police uniform, standing in what appears to be an indoor setting with a brown door and a tiled floor in the background…"

**Line1**: "After all, tomorrow is another day."

**Line2**: "May the Force be with you."

**Line3**: "Now…Say my name."

**Line4**: "I'll be back."

Figure S9: More demo results on audio-visual subject customization.

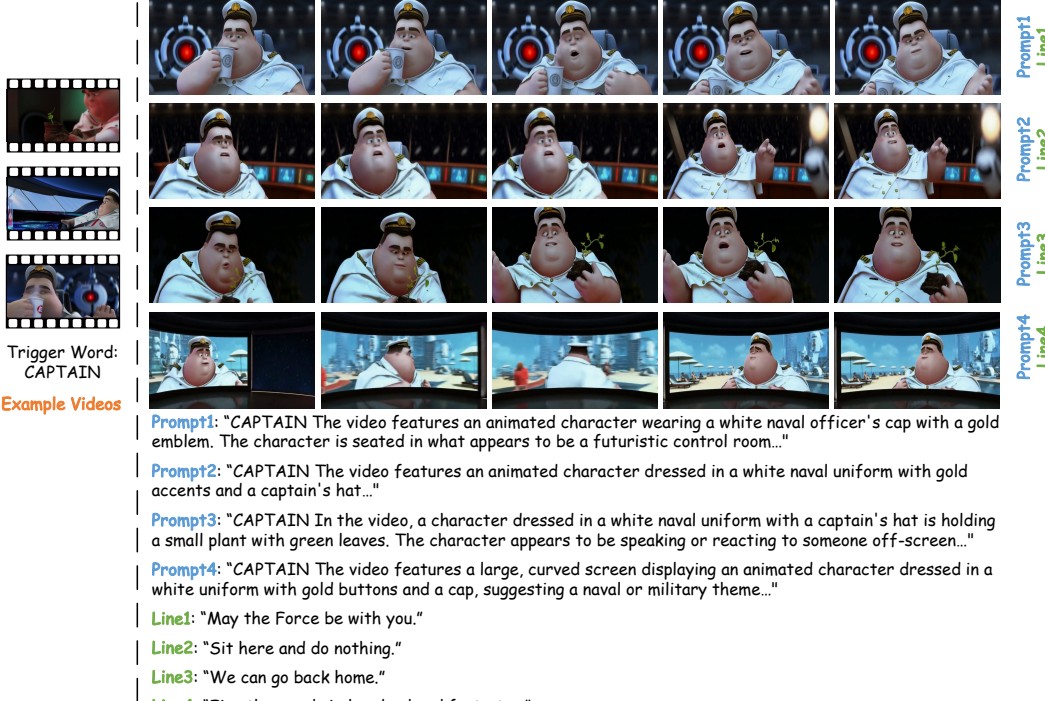

Trigger Word:
CAPTAIN

Example Videos

**Prompt1**: "CAPTAIN The video features an animated character wearing a white naval officer's cap with a gold emblem. The character is seated in what appears to be a futuristic control room…"

**Prompt2**: "CAPTAIN The video features an animated character dressed in a white naval uniform with gold accents and a captain's hat…"

**Prompt3**: "CAPTAIN In the video, a character dressed in a white naval uniform with a captain's hat is holding a small plant with green leaves. The character appears to be speaking or reacting to someone off-screen…"

**Prompt4**: "CAPTAIN The video features a large, curved screen displaying an animated character dressed in a white uniform with gold buttons and a cap, suggesting a naval or military theme…"

**Line1**: "May the Force be with you."

**Line2**: "Sit here and do nothing."

**Line3**: "We can go back home."

**Line4**: "Five thousand six hundred and forty-two"

Figure S10: More demo results on audio-visual subject customization.

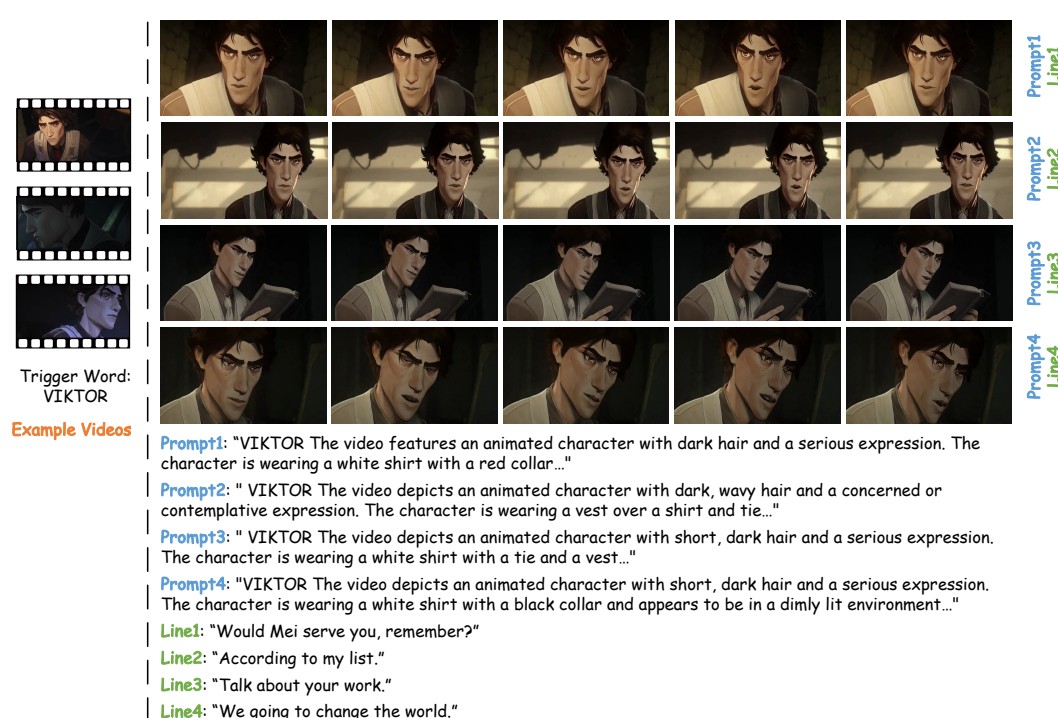

Trigger Word:
VIKTOR

Example Videos

Prompt1: "VIKTOR The video features an animated character with dark hair and a serious expression. The character is wearing a white shirt with a red collar…"

Prompt2: " VIKTOR The video depicts an animated character with dark, wavy hair and a concerned or contemplative expression. The character is wearing a vest over a shirt and tie…"

Prompt3: " VIKTOR The video depicts an animated character with short, dark hair and a serious expression. The character is wearing a white shirt with a tie and a vest…"

Prompt4: "VIKTOR The video depicts an animated character with short, dark hair and a serious expression. The character is wearing a white shirt with a black collar and appears to be in a dimly lit environment…"

Line1: "Would Mei serve you, remember?"

Line2: "According to my list."

Line3: "Talk about your work."

Line4: "We going to change the world."

Figure S11: More demo results on audio-visual subject customization.

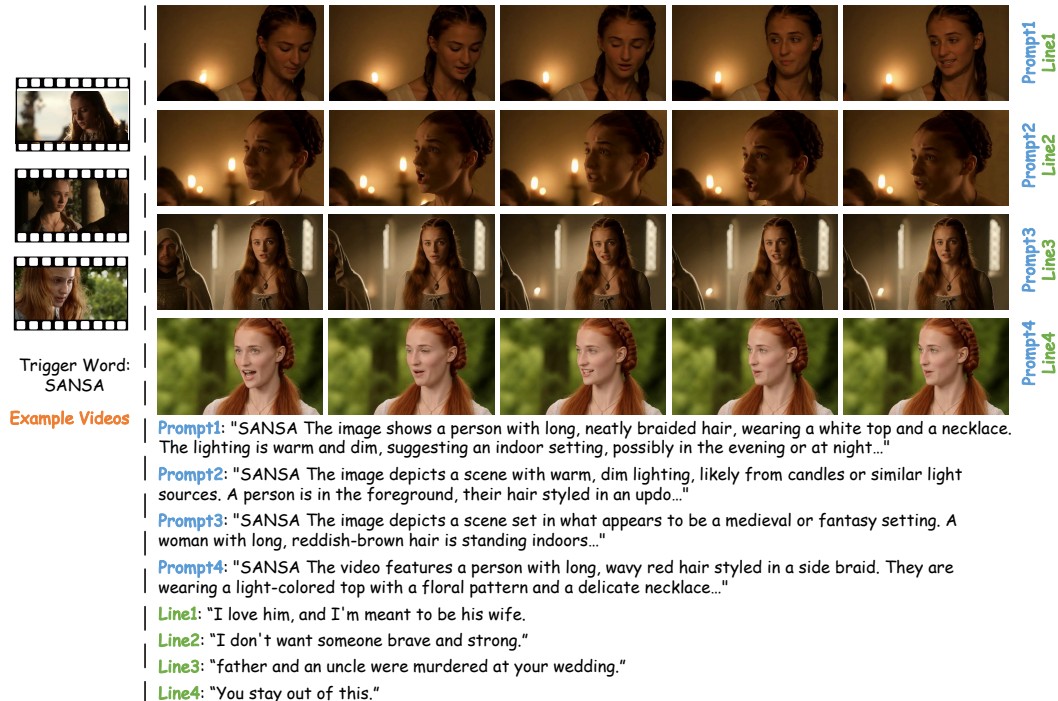

Trigger Word:
SANSA

Example Videos

Prompt1: "SANSA The image shows a person with long, neatly braided hair, wearing a white top and a necklace. The lighting is warm and dim, suggesting an indoor setting, possibly in the evening or at night…"

Prompt2: "SANSA The image depicts a scene with warm, dim lighting, likely from candles or similar light sources. A person is in the foreground, their hair styled in an updo…"

Prompt3: "SANSA The image depicts a scene set in what appears to be a medieval or fantasy setting. A woman with long, reddish-brown hair is standing indoors…"

Prompt4: "SANSA The video features a person with long, wavy red hair styled in a side braid. They are wearing a light-colored top with a floral pattern and a delicate necklace…"

Line1: "I love him, and I'm meant to be his wife.

Line2: "I don't want someone brave and strong."

Line3: "father and an uncle were murdered at your wedding."

Line4: "You stay out of this."

Figure S12: More demo results on audio-visual subject customization.

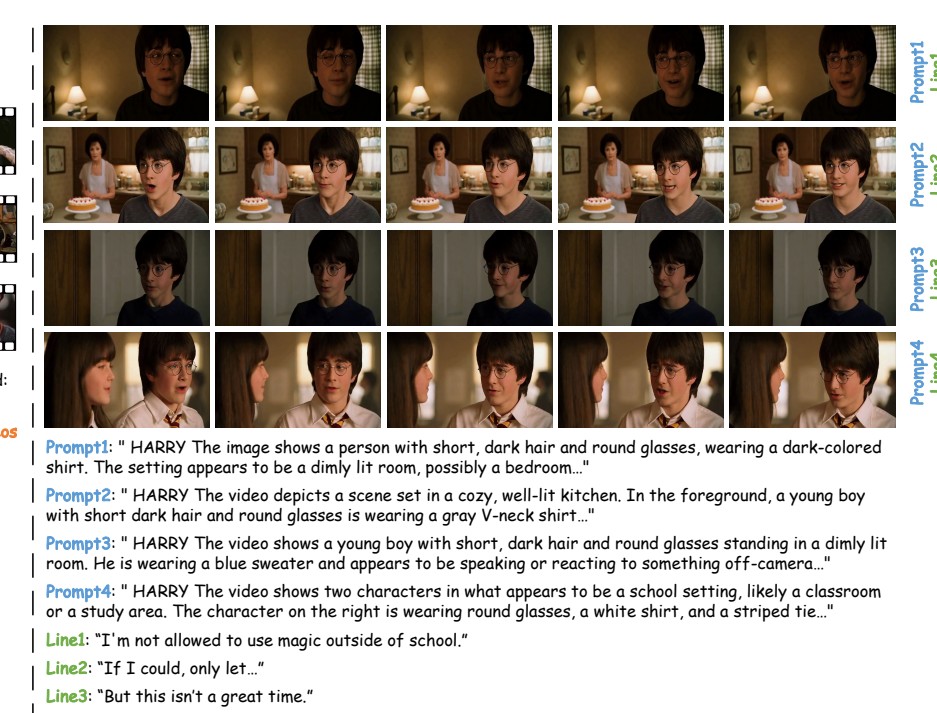

Trigger Word:
HARRY

Example Videos

**Prompt1:** " HARRY The image shows a person with short, dark hair and round glasses, wearing a dark-colored shirt. The setting appears to be a dimly lit room, possibly a bedroom…"

**Prompt2:** " HARRY The video depicts a scene set in a cozy, well-lit kitchen. In the foreground, a young boy with short dark hair and round glasses is wearing a gray V-neck shirt…"

**Prompt3:** " HARRY The video shows a young boy with short, dark hair and round glasses standing in a dimly lit room. He is wearing a blue sweater and appears to be speaking or reacting to something off-camera…"

**Prompt4:** " HARRY The video shows two characters in what appears to be a school setting, likely a classroom or a study area. The character on the right is wearing round glasses, a white shirt, and a striped tie…"

**Line1:** "I'm not allowed to use magic outside of school."

**Line2:** "If I could, only let…"

**Line3:** "But this isn't a great time."

**Line4:** "That's what's guarding. That's what's Snape want."

Figure S13: More demo results on audio-visual subject customization.

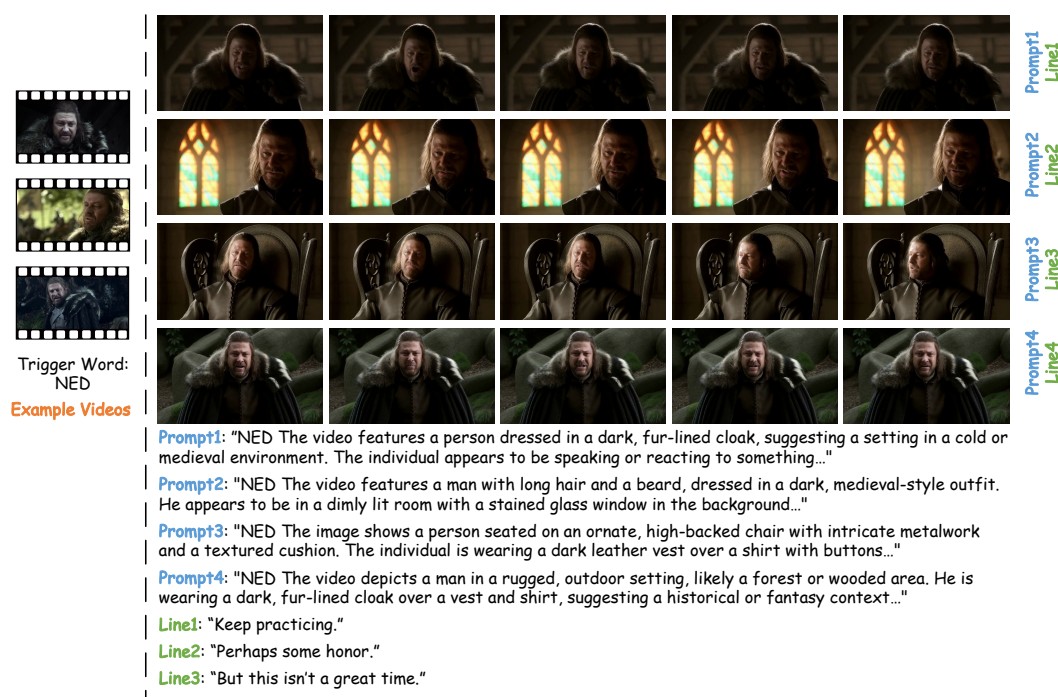

Trigger Word:
NED

Example Videos

**Prompt1:** "NED The video features a person dressed in a dark, fur-lined cloak, suggesting a setting in a cold or medieval environment. The individual appears to be speaking or reacting to something…"

**Prompt2:** "NED The video features a man with long hair and a beard, dressed in a dark, medieval-style outfit. He appears to be in a dimly lit room with a stained glass window in the background…"

**Prompt3:** "NED The image shows a person seated on an ornate, high-backed chair with intricate metalwork and a textured cushion. The individual is wearing a dark leather vest over a shirt with buttons…"

**Prompt4:** "NED The video depicts a man in a rugged, outdoor setting, likely a forest or wooded area. He is wearing a dark, fur-lined cloak over a vest and shirt, suggesting a historical or fantasy context…"

**Line1:** "Keep practicing."

**Line2:** "Perhaps some honor."

**Line3:** "But this isn't a great time."

**Line4:** "Sentence you to die. You'll bury him yourself."

Figure S14: More demo results on audio-visual subject customization.

