# OpenReview forum: "From Silence to Sound: Towards Audio-Visual Subject Customization"
_ICLR.cc/2026/Conference — ICLR 2026 Conference Withdrawn Submission_

### Official Review · Reviewer_svjj · 2025-10-31

**Soundness:** 3
**Presentation:** 3
**Contribution:** 2
**Rating:** 4
**Confidence:** 3

**Summary:**

The paper proposes a new task, audio-visual subject customization, as well as a new system, VauCustom, a two-stage pipeline (zero-shot TTS → audio-conditioned T2V) with a four-step training schedule (dataset LoRA->audio cross-attn->subject LoRA->joint finetune) plus region-selective audio CFG. On a 60-identity benchmark including (human, animated human and animated animal), they compare against several prior baselines in metrics in terms of both semantic alignment and audio-visual synchronization; qualitative/quantitative results against portrait-animation baselines favor the method.

**Strengths:**

1. The paper is mostly clear, and with straightforward pipeline illustrations.

2. Region-selective audio CFG is a neat and novel inference-time solution that improves lip-sync while limiting background artifacts; and visualizations in the attention mask part is very intuitive.

3. The experiments part is exhaustive, and it includes detailed main experiments and ablation studies. The metrics cover both “semantic” (Text/Video CLIP), identity (DINO), dynamics/consistency, and sync (Sync-C/D). Audio quality metrics (SIM-o, WER, UTMOS) are provided as shown in supplementary material.

**Weaknesses:**

1. The main concern for me is the complication of 4 sub-stage training for audio-conditioned video generation part, it seems that in order to adapt to each character, a character specific lora (sub staged) must be finetuned and then jointly tune the audio cross-attention part (sub stage 4). What's the typical finetuning time for each new character in order to get a good enough checkpoint for sampling?

2. The paper should compare prior works on audio-condition video generation methods (e.g., AudCast (also missing citation since very relevant) ) and joint text to video audio generation (different variants of AV-DiTs such as JarvisDiT).

3. The scope of the paper is still limited to talking head portrait animation scenario, which I think existing research is pretty much well-established in this domain. To really showcase the capability of the method, the paper needs to demonstrate its capbility not only just control the face part of the portait, but the body motions and even background. Though I understand that this is out of scope for this paper, this will mark more convincing achievements.

4. How does the proposed method generalize to out of domain in-the-wild videos with strovnger visual variations in both foreground facial region and background area, I think a small eval set to test its in-the-wild generalization capability will bring more impact to the method.

**Questions:**

See weaknesses.

---

### Official Review · Reviewer_zYdy · 2025-11-02

**Soundness:** 2
**Presentation:** 2
**Contribution:** 2
**Rating:** 4
**Confidence:** 4

**Summary:**

The paper introduced a new audio-visual subject customization task that generates videos featuring user-defined characters, emphasizing both visual and audio dimensions. A key challenge is mitigating the gap between visual synthesis and audio learning. To tackle this, it proposed VauCustom (Video-Audio Custom), a two-stage method that leverages zero-shot text-to-speech to create personalized audio, and then conditions video synthesis on this audio to unify audio and visuals. The decoupled learning strategy intelligently handles the challenge of integrating audio control into pre-trained visual models without sacrificing appearance fidelity, while the region-selective CFG solves the artifact problem associated with global audio conditioning. Experiments demonstrate that VauCustom delivers consistent character appearance, natural audio quality, and precise audio-video synchronization across diverse scenarios, including real humans, animated human  characters, and animal characters.

**Strengths:**

+ Define the audio-visual subject customization task and establish the first benchmark,  featuring diverse subjects such as real humans, animated humans, and animal-style characters, along with standardized metrics for appearance, audio quality, and synchronization.

+ It presents VauCustom, a two-stage framework integrating zero-shot TTS audio with audio-conditioned video synthesis to achieve synchronized audio visual customization.

+ It introduced tailored strategies, including a decoupled audio-visual learning method to preserve visual fidelity and a region-selective audio CFG mechanism to improve lip–audio synchronization, leading to superior results across diverse scenarios.

**Weaknesses:**

-- The proposed task is interesting but its real-world application scenarios should be more clear.

-- The proposed methods are incremental. The two-stage pipeline relies heavily on the zero-shot TTS model for the initial audio generation. Any limitation or artifact in the generated audio will inevitably cascade as a performance ceiling for the final video synthesis and synchronization. The paper notes the use of an open-source TTS model, but does not extensively ablate the impact of different TTS quality on final sync.

-- The audio quality evaluation is briefly presented in Table S1 but is only for the synthesized audio and not a direct comparison against baselines like SadTalker, Hallo3, or AniPortrait. Since the audio is generated independently in Stage 1 using Cosy Voice2, the excellent audio metrics reflect the quality of the TTS model, not the end-to-end framework's unique contribution in a comparative sense.

-- The pipeline relies heavily on the zero-shot TTS model for the initial audio generation. Any limitation or artifact in the generated audio (Speaker Similarity) will inevitably cascade as a performance ceiling for the final video synthesis. The paper notes the use of an open-source TTS model, but does not extensively ablate the impact of different TTS quality on final sync.

**Questions:**

Please address my major concerns as listed in the weakness section.

**Details Of Ethics Concerns:**

The primary ethical concern is the paper's contribution to technologies capable of generating highly realistic, customized videos featuring user-defined characters with synchronized speech. This capability inherently carries the risk of misuse for creating misleading or harmful content, specifically deepfakes.

---

### Official Review · Reviewer_VGNM · 2025-11-03

**Soundness:** 3
**Presentation:** 2
**Contribution:** 3
**Rating:** 4
**Confidence:** 4

**Summary:**

This paper introduces the novel task of audio-visual subject customization: generating a short video clip of a character along with their speech, conditioned on a visual scene text description, a dialog line and a set of exemplar clips from the aforementioned character.

The paper claims the following contributions:
- The definition of the task along with the release of a standard evaluation benchmark. The proposed comprehensive benchmark covering real humans, animated human characters, and animal-style characters, along with standardized evaluation protocols on three dimensions: character appearance, audio quality, and audio-visual synchronization.
- The VauCustom framework, a two-stage framework involving zero-shot text-to-speech followed by video-generation conditioned on generated audio to achieve synchronized audiovisual customization.
- Two model training and inference strategies (decoupled audio-visual learning, region-selective audio CFG mechanism) meant to preserve visual fidelity and improve lip-audio synchronization.

Compared with previous state-of-the art, this paper addresses the overlooked aspect of character speech, including both lip movements and synchronized audio.

The video generation DiT model (pretrained WAN) is conditioned on both the pre-generated audio of the target character and a text description of the video. The model is finetuned in 4 stages:
- Appearance LoRA on talking face dataset.
- Audio cross attention on talking face dataset.
- Customize appearance LoRA on reference videos of the target character.
- Joint audio cross attention and appearance finetuning on reference videos.

The decoupled training strategy is meant to prevent the audio conditioning modules to inadvertently learn appearance-specific biases.

The inference pipeline first employs the pretrained CosyVoice 2 model to generate the target speech sample conditioned on the dialogue line and an expressivity prompt. The the video generation model generates the video clip conditioned on the audio and a text description of the scene.

The proposed benchmark consists in 60 characters, 10 clips per character for training, 8 for evaluation. 480 test samples total.

The proposed VauCustom framework is compared to SadTalker, AniPortrait and Hallo3 (visual-only models that do not support audiovisual alignment). On the proposed benchmark, VauCustom outperforms the baselines across the board. It notably shines for motion dynamics (much more dynamic scenes than baselines) and audio-visual synchronization (thanks to the audio-conditioning focus of this paper).

Overall the novelty and significance of this paper make it a good candidate for acceptance. However the paper's quality is hindered by several lacks of details and experiments supporting the design choices (audio tokens, trigger word?).

**Strengths:**

The main strength of the paper in originality: the definition of a novel task along with the proposal of a related benchmark.

The proposed method is well supported by several qualitative examples that also include baseline systems.

**Weaknesses:**

The proposed method requires more than two stages. The video generation model requires two generic finetuning stages followed by two subject-specific finetuning stages on the exemplar clips of the target character (visual only, then joint).

Several details are missing to enable reproducibility.
- The authors claim they will release code, data and models but as of today this is not the case.
- Some details are missing regarding the different finetuning stages (learning rates, batch size...).
- The binarization parameters of the cross attention maps are missing.
- Some design choices lack justification or experimental support (choice of trigger words, learnable additional audio tokens, cfg ablations).

**Questions:**

- What is the exact role and design of the learnable audio tokens in the stage 3.d?
- What does “vector” mean in the Figure 4?
- What are the binarization parameters in the Figure 4? How are they chosen?
- Why are the cross attention map activations taken from the interaction with a text trigger word instead of the audio token projections?
- As CFG is a core contribution of this paper, I would expect some CFG ablation curves somewhere in the paper.
- Are the last two finetuning stages run independently on each target character data (creating one finetuned model per character) or on the whole set of 60 characters?
- How does the model behave on out of distribution (unseen) characters?

Typo in the Table 1 ("consistency").

---

### Note · Authors · 2025-11-13

I have read and agree with the venue's withdrawal policy on behalf of myself and my co-authors.